# The rate and molecular spectrum of mutation are selectively maintained in yeast

Haoxuan Liu[1] & Jianzhi Zhang [1✉]

What determines the rate ($\mu$) and molecular spectrum of mutation is a fundamental question. The prevailing hypothesis asserts that natural selection against deleterious mutations has pushed $\mu$ to the minimum achievable in the presence of genetic drift, or the drift barrier. Here we show that, contrasting this hypothesis, $\mu$ substantially exceeds the drift barrier in diverse organisms. Random mutation accumulation (MA) in yeast frequently reduces $\mu$, and deleting the newly discovered mutator gene *PSP2* nearly halves $\mu$. These results, along with a comparison between the MA and natural yeast strains, demonstrate that $\mu$ is maintained above the drift barrier by stabilizing selection. Similar comparisons show that the mutation spectrum such as the universal AT mutational bias is not intrinsic but has been selectively preserved. These findings blur the separation of mutation from selection as distinct evolutionary forces but open the door to alleviating mutagenesis in various organisms by genome editing.

[1] Department of Ecology and Evolutionary Biology, University of Michigan, Ann Arbor, MI, USA. ✉email: jianzhi@umich.edu

Mutation is the ultimate source of all genetic variations, including those driving adaptations and those causing hereditary diseases. Therefore, the mutation rate per nucleotide per generation ($\mu$) and its evolution are of broad relevance and interest. Because the vast majority of mutations are deleterious, Sturtevant famously asked in 1937 why $\mu$ has not been reduced by natural selection to zero[1]. While he sighed that "no answer seems possible at present", much progress has been made in the intervening years[2–13]. It is now recognized that an organism's $\mu$ is jointly determined by its genotype[14] and environment[15] and is subject to natural selection[6,12], and that the selection can arise from three factors: deleterious mutations, beneficial mutations, and the cost of fidelity[6,13]. Deleterious mutations reduce organismal fitness, leading to the selective fixations of mutation rate modifiers that lower $\mu$ and a decrease of $\mu$ (Fig. 1a). By contrast, beneficial mutations raise organismal fitness, leading to the selective fixations of mutation rate modifiers that increase $\mu$ and an elevation of $\mu$ (Fig. 1a). In these two selections, the fitness effect of the modifier lies entirely in the mutations created and linked with the modifier, so the modifier is subject only to the so-called second-order selection. The cost of fidelity refers to the fitness cost due to the energy and time spent on proofreading, repair, and other biological processes that reduce $\mu$. Hence, the cost of fidelity creates a first-order selection for mutation rate modifiers that increase $\mu$, resulting in an uplift of $\mu$ (Fig. 1a). Therefore, a nonzero $\mu$ can result from a balance between the respective selections for lower and higher $\mu$. We will refer to this answer to Sturtevant's question as the conventional model.

In the last decade, however, an alternative model termed the drift–barrier hypothesis (DBH) has emerged as the prevailing hypothesis of mutation rate evolution[12]. The DBH considers only the first of the three selections aforementioned and regards the selections for higher $\mu$ negligible. The DBH posits that, as $\mu$ diminishes, the selective benefit of a given fractional reduction of $\mu$ also diminishes; eventually, the benefit becomes so weak relative to genetic drift that $\mu$ can no longer descend, especially when mutations are biased toward creating modifiers that increase $\mu$.

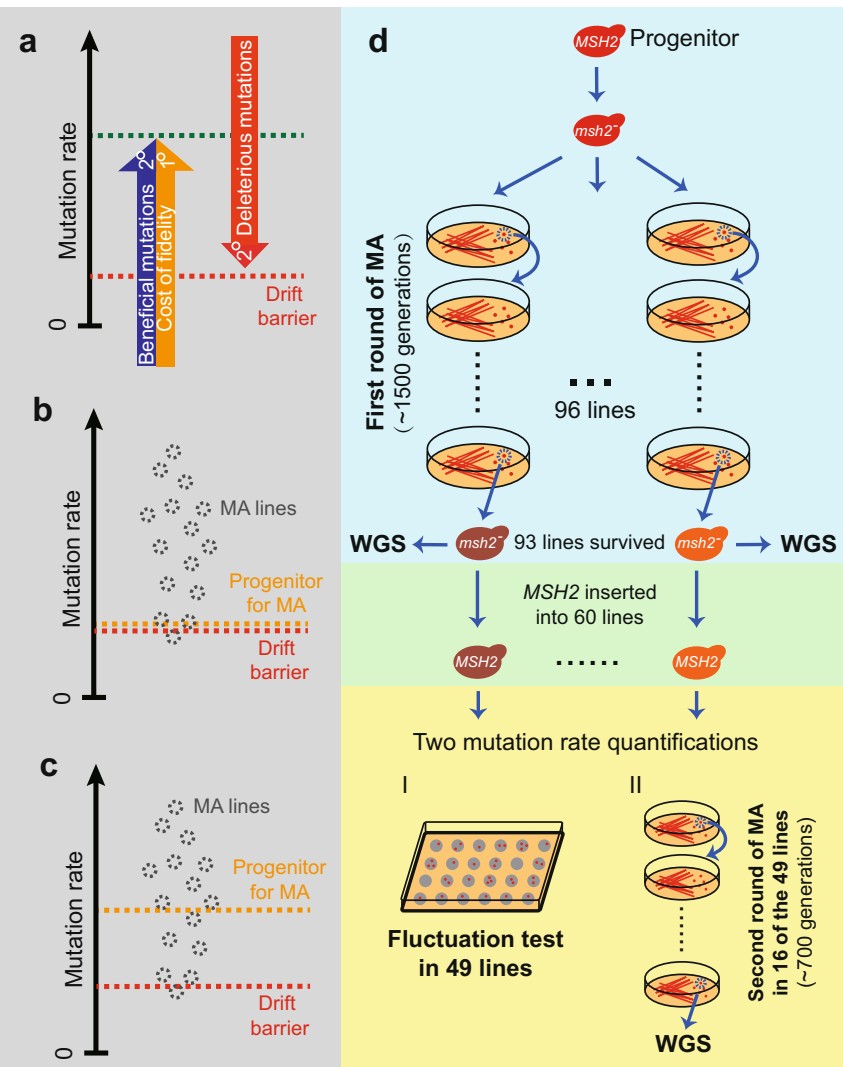

**Fig. 1 Theoretical framework of mutation rate evolution and the study design. a** Three selective forces (arrows) potentially drive the evolution of the mutation rate $\mu$. 1° and 2° represent first-order and second-order selections, respectively. Beneficial mutations and cost of fidelity induce selections for higher $\mu$, while deleterious mutations induce a selection for lower $\mu$. If the selections raising $\mu$ are negligible, $\mu$ is pushed to the red dotted line that represents the drift barrier; otherwise, $\mu$ is at the green dotted line that is well above the drift barrier. Predicted relationships in $\mu$ among the drift barrier (red line), progenitor for mutation accumulation (MA) (orange line), and multiple resultant MA lines (gray circles), under the drift barrier hypothesis (**b**) or the conventional model in which a balance between opposing selections maintains $\mu$ well above the drift barrier (**c**). **d** Study design. WGS, whole-genome sequencing.

The minimal $\mu$ achievable by selection against deleterious mutations in the presence of drift is known as the drift barrier.

Which of these competing hypotheses provides the right answer to Sturtevant's question? While a key prediction of the DBH has been validated (see "Discussion"), it is unknown whether the selections for higher $\mu$, which have received empirical support (see "Discussion"), are so inconsequential that $\mu$ approaches the drift barrier. Because $\mu$ would be subject to stabilizing selection under the conventional model but directional selection under the DBH, it is possible to distinguish between them by evaluating the type of selection acting on $\mu$. Another key aspect of mutation is its molecular spectrum, defined by the relative rates of different mutation types. Mutation spectrum could affect the severity of mutations[16] and influence adaptation[17], but whether the mutation spectrum itself is subject to natural selection is unknown.

In this work, we use the budding yeast *Saccharomyces cerevisiae* as a model to assess selections in the evolution of mutation rate and spectrum. We show that yeast's mutation rate is maintained well above the drift barrier by stabilizing selection and that its mutation spectrum has also been shaped by selection.

## Results

**Mutagenesis frequently reduces $\mu$.** The type of selection acting on a trait can be inferred from its phenotypic change upon the removal of the selection. If $\mu$ has been selectively minimized to the drift barrier, evolution in the absence of selection should generally cause a rise in $\mu$, although occasional small reductions in $\mu$ cannot be excluded[18] (Fig. 1b). By contrast, if the mutation rate is selectively maintained well above the drift barrier, upon the removal of selection, the probabilities for $\mu$ to go up and go down are both substantial (Fig. 1c). Following this logic, we initiated 96 mutation accumulation (MA) lines from a commonly used laboratory yeast strain, in which the mismatch repair gene *MSH2* had been deleted to speed up the accumulation of mutations (see "Methods"). Each MA line went through on average 1511 mitotic generations including 80 evenly spaced single-cell bottlenecks to minimize the effective population size ($N_e$) and selection; 93 lines survived the MA (Fig. 1d).

As expected, yeast growth significantly slowed after the MA; the average growth rate dropped by 41% (Supplementary Fig. 1a). Whole-genome sequencing (WGS) showed that, on average, each MA line accumulated 879 mutations, including 115 single nucleotide variants (SNVs) and 764 insertions/deletions (indels) (see "Methods"), consistent with the known mutation spectrum of *MSH2*-lacking strains[19]. An average of 186 genes was hit by mutations per MA line (Supplementary Fig. 1b, Supplementary Table 1, Supplementary Data 1). Consistent with the report that *MSH2*-lacking strains suffer from increased rates of indel mutations in homonucleotide runs[20], 78% of mutations in our MA lines were shared by at least two lines and almost all of the shared mutations were indels located in homonucleotide runs (see "Methods").

Because the $N_e$ of the MA lines was about 10 (see "Methods") while most mutations are expected to have a fitness effect on the order of 1% or smaller[21], selection should be infrequent during the MA and was indeed the case (see "Methods"). To assess the impact of MA on $\mu$, we first inserted *MSH2* back to the MA lines, which was accomplished in 60 of the 93 lines (Fig. 1d). Using the classic fluctuation test based on the reporter gene *CAN1* (see "Methods"), we successfully measured $\mu$ in the progenitor as well as 49 of the above 60 MA lines, all carrying an intact *MSH2* (Fig. 1d). We subsequently found that five of the 49 lines were likely diploidized upon the insertion of *MSH2* (see the following paragraph) and excluded them (marked with stars in Fig. 2a) from all *CAN1*-based analyses. We found $\mu$ of the 44 remaining MA lines to range from 0.01 to 26 times that of the progenitor (Fig. 2a, Supplementary Data 1), including 19 lines with significantly higher $\mu$ and 13 lines

with significantly lower $\mu$ than the progenitor (see "Methods"). Furthermore, 10 of the 13 lines with significantly decreased $\mu$ had $\mu$ reduced by at least 50%, while the remaining three lines had $\mu$ reduced by 40% to 43% (Fig. 2a). That over 40% of MA lines with significantly altered $\mu$ exhibit such drastic reductions in $\mu$ is inconsistent with the DBH, because when $\mu$ is near the drift barrier, mutations are expected to be strongly biased toward increasing $\mu$ and are not expected to cause such large reductions of $\mu$ so frequently (Fig. 1b). Our finding suggests that the progenitor's $\mu$ is well above the drift barrier (Fig. 1c). We found a significant positive correlation between $\mu$ and the number of mutations accumulated during the MA, but $\mu$ was not significantly correlated with the growth rate of the MA line (Supplementary Fig. 2).

Because the above estimation of $\mu$ was based on loss-of-function mutations in one gene, we attempted to verify these results by performing another round of MA followed by WGS in 16 of the above 49 MA lines as well as the progenitor (and a diploid version of the progenitor), all with an intact *MSH2* (Fig. 1d). 4–20 parallel lines were established from each strain, and on average 684 generations of MA were performed in the medium similar to that used in the fluctuation test (Supplementary Table 2, Supplementary Data 1, see "Methods"). Four of the 16 MA lines were apparently diploid, because the majority of the mutations observed in MA + WGS were in heterozygous state. Diploids should not produce mutant colonies in the fluctuation test. To be conservative in inferring mutation rate reductions in the first round of MA, we additionally regarded a line that was not subject to MA + WGS but had only two mutant colonies in the fluctuation test as putatively diploid (right most line marked with a star in Fig. 2a). We excluded these five diploid lines from all *CAN1*-based analyses. Because haploid and diploid progenitors showed similar mutation rates (Fig. 2b, c), all 16 lines with MA + WGS were included in analyses based solely on MA + WGS. The MA + WGS results were generally consistent with those from the fluctuation test. For instance, compared with the progenitor, all eight lines with higher *CAN1*-based $\mu$ exhibited higher MA + WGS-based SNV (Fig. 2b) or indel (Fig. 2c) rates. Among the four lines with lower *CAN1*-based $\mu$, three exhibited significantly lower MA + WGS-based SNV or indel rates (Fig. 2a). *CAN1*-based $\mu$ is significantly correlated with both the MA + WGS-based SNV rate ($r = 0.88$, $P = 8.9 \times 10^{-5}$; Fig. 2d) and the MA + WGS-based indel rate ($r = 0.78$, $P = 1.6 \times 10^{-3}$; Fig. 2e), although the latter correlation is weaker than the former. This observation is not unexpected given that most loss-of-function mutations in *CAN1* are SNVs instead of indels[22].

**Stabilizing selection of $\mu$.** The above analysis strongly suggests that $\mu$ is not selectively minimized to the drift barrier in the progenitor. To assess the selective forces acting on $\mu$, we took advantage of published *CAN1*-based $\mu$ estimates from seven natural yeast strains of diverse origins[14] (Supplementary Table 3, Supplementary Fig. 3). For a neutrally evolving trait, the ratio of its genetic variance among natural strains of a species ($V_g$) to the mutational variance generated by mutations per generation ($V_m$) is expected to equal $4N_e$ in primarily asexual diploids such as *S. cerevisiae*[23]. By contrast, stabilizing selection would reduce $V_g$ and render $V_g/V_m$ smaller than $4N_e$. Because $\mu$ is not normally distributed among the MA lines, we first $\log_{10}$-transformed $\mu$ (Fig. 2a) before computing $V_m$ and $V_g$, although the results are not qualitatively different without the transformation (Supplementary Table 4). We estimated $V_g$ of $\mu$ from the seven natural strains. To estimate $V_m$ that is comparable with $V_g$, we used the *CAN1*-based $\mu$ estimates from the 44 haploid MA lines, but corrected for the increased mutagenesis in the MA induced by deleting *MSH2*. We employed three corrections by respectively assuming that deleting *MSH2* caused the same fold change in the rate of each mutation type as the observed fold change of the total rate of

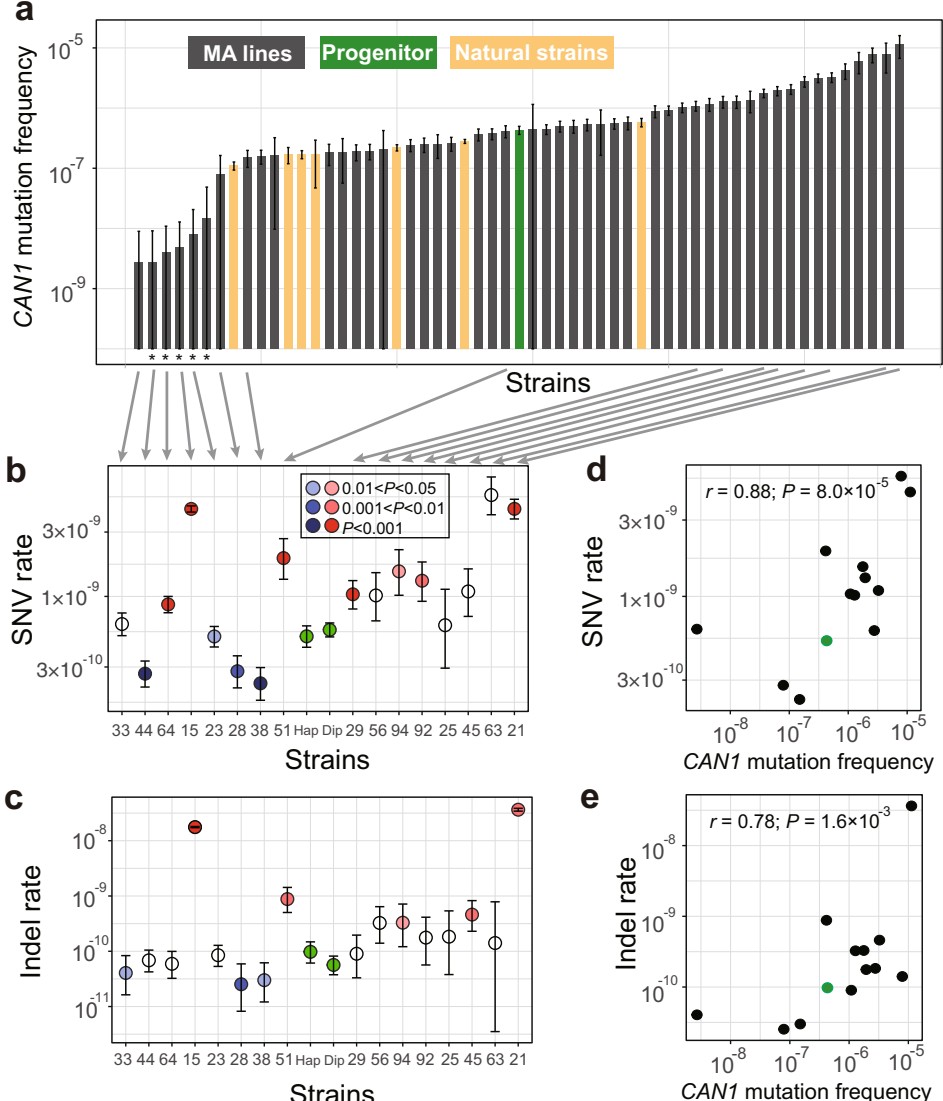

**Fig. 2 Mutation frequencies and rates of the MA lines. a** *CAN1* mutation frequencies of the progenitor (green), 49 MA lines (gray), and 7 natural yeast strains (yellow), determined by the fluctuation test based on 72 (green and gray) or 288 (yellow) biologically independent cell cultures from each strain. The data from the seven natural strains came from ref. [14]. Error bars indicate 95% confidence intervals of the mean. "*" on the X-axis indicates diploid or putatively diploid strains. Mutation frequency is the probability of loss-of-function mutation in *CAN1* per cell division, so is not directly comparable with the mutation rates estimated by MA + WGS. SNV (**b**) or indel (**c**) mutation rate per site per generation in the progenitor in both haploid and diploid form (green) and 16 MA lines estimated by MA + WGS based on 4 to 20 biologically independent replicates. Numbers on the X-axis refer to IDs of MA lines, while "Hap" refers to the progenitor in the haploid form and "Dip" refers to the progenitor in the diploid form. Circles represent mean values while error bars show 95% confidence intervals predicted from Poisson distributions. Significant rate differences from the progenitor are indicated by blue (lower than the progenitor) or red (higher than the progenitor) with different shades for different nominal *P* values from Wilcoxon rank-sum tests. White circles show no significant rate difference from the progenitor. Correlation between SNV (**d**) or indel (**e**) mutation rate measured by MA + WGS and *CAN1* mutation frequency measured in fluctuation test among the 12 haploid MA lines and the haploid progenitor (green). Pearson's *r* (based on the values before the log₁₀-transformation) and associated *P* value are shown. The green dot shows the progenitor.

SNVs and indels ($V_{m1}$), as that of indels ($V_{m2}$), and as that of SNVs ($V_{m3}$). Because deleting *MSH2* increased the indel rate much more than increasing the SNV rate (see "Methods"), among the three $V_m$ values, $V_{m1}$ is probably the closest to the truth, while $V_{m2}$ is underestimated and $V_{m3}$ is overestimated. Hence, $V_{m3}$ and $V_{m2}$ allow determining the lower and upper bounds of $V_g/V_m$, respectively.

We found $V_g/V_m$ to be at least 540 times lower than the neutral expectation of $4N_e \approx 4 \times 10^7$ (see "Methods") (Table 1), regardless of the particular $V_m$ used (Table 1), indicating strong stabilizing selection of $\mu$. This signal of stabilizing selection is not an artifact of the physical limits of $\mu$, because the range of $\mu$ among the

natural strains is even smaller than that of the MA lines (Fig. 2a). To investigate whether the stabilizing selection prohibits the evolution of higher $\mu$, lower $\mu$, or both, we separated $V_m$ into two components that respectively measure the variance of $\mu$ created by mutations decreasing $\mu$ ($V_{mL}$) and increasing $\mu$ ($V_{mH}$). If there is no selection against a reduction in $\mu$, $V_g$ should be at least as large as $4N_e V_{mL}$. However, we found $V_g$ to be at least 300 times lower than $4N_e V_{mL}$ (Table 1), indicating the action of selection prohibiting a reduction of $\mu$ in evolution. Similarly, $V_g$ was at least 230 times lower than $4N_e V_{mH}$ (Table 1), indicating the action of selection prohibiting a rise of $\mu$ in evolution. In the above tests,

**Table 1 Test of stabilizing selection of the mutation rate in yeast.**

| | $V_m$ | | | $V_{mL}$ | | | $V_{mH}$ | | |
|---|---|---|---|---|---|---|---|---|---|
| | $V_{m1}$ | $V_{m2}$ | $V_{m3}$ | $V_{mL1}$ | $V_{mL2}$ | $V_{mL3}$ | $V_{mH1}$ | $V_{mH2}$ | $V_{mH3}$ |
| *CAN1*-based tests ($V_g$: $4.5 \times 10^{-2}$) | | | | | | | | | |
| Mutational variance | $3.4 \times 10^{-6}$ | $6.1 \times 10^{-7}$ | $2.2 \times 10^{-5}$ | $1.9 \times 10^{-6}$ | $3.4 \times 10^{-7}$ | $1.2 \times 10^{-5}$ | $1.4 \times 10^{-6}$ | $2.6 \times 10^{-7}$ | $9.2 \times 10^{-6}$ |
| $V_g/V_m$ (neutral expectation: $4 \times 10^7$) | $1.3 \times 10^4$ ¶ | $7.4 \times 10^4$ ¶ | $2.0 \times 10^3$ ¶ | $2.4 \times 10^4$ ¶ | $1.3 \times 10^5$ ¶ | $3.7 \times 10^3$ ¶ | $3.1 \times 10^4$ ¶ | $1.7 \times 10^5$ ¶ | $4.9 \times 10^3$ ¶ |
| MA+WGS-based tests ($D^2$: 0.18) | | | | | | | | | |
| Mutational variance | $1.4 \times 10^{-6}$ | $2.6 \times 10^{-7}$ | $9.3 \times 10^{-6}$ | $1.3 \times 10^{-7}$ | $2.6 \times 10^{-8}$ | $9.3 \times 10^{-7}$ | $8.5 \times 10^{-7}$ | $1.5 \times 10^{-7}$ | $5.5 \times 10^{-6}$ |
| $D^2/V_m$ (neutral expectation: $2.89 \times 10^9$) | $1.3 \times 10^5$ ¶ | $6.9 \times 10^5$ ¶ | $1.9 \times 10^4$ ¶ | $1.3 \times 10^6$ * | $7.0 \times 10^6$ * | $1.9 \times 10^5$ * | $2.1 \times 10^5$ ¶ | $1.2 \times 10^6$ ¶ | $3.3 \times 10^4$ ¶ |

All mutation frequencies/rates are log$_{10}$-transofrmed before the test. *CAN1*-based intraspecific mutation rate variance $V_g$ is from 7 natural strains while $V_m$ is from 44 haploid MA lines. MA+WGS-based $D^2$ is the squared difference in SNV mutation rate between *S. cerevisiae* and *S. paradoxus*, while $V_m$ is based on the SNV mutation rates of 16 MA lines. All $V_g/V_m$ and $D^2/V_m$ ratios are significantly below the corresponding neutral expectations based on bootstrap tests (*, $P < 0.001$; ¶, $P < 0.0001$).

the smallest difference observed between $V_g$ and a neutral expectation was 230 times, based on $V_{m2}$ that corresponds to a conservative test. Therefore, it is exceedingly unlikely that our test results are due to confounding factors such as mutation spectrum differences between wild-type and *MSH2*-lacking strains or the inaccuracies of $V_m$, $V_g$ and $N_e$ estimates (see "Methods"). Together, the above results demonstrate that $\mu$ has been selectively maintained at an intermediate level in *S. cerevisiae* Note that the selective forces to increase and to suppress $\mu$ are not equally strong, because the mean $\mu$ of the MA lines is higher than the progenitor ($P = 4.3 \times 10^{-3}$ for *CAN1*-based $\mu$, *t*-test; $P = 5.7 \times 10^{-7}$ for WGS-based SNV rate, *t*-test).

To examine if the above finding extends beyond the species concerned, we examined the evolution of $\mu$ in the divergence between *S. cerevisiae* and its sister species *S. paradoxus*. *S. paradoxus*' SNV rate was recently estimated by MA + WGS to be $7.27 \times 10^{-11}$ per site per generation[24], about one third that of *S. cerevisiae*[15]. Under neutral evolution, the squared difference in mutation rate between the two species ($D^2$) should equal $V_m$ times the number of generations separating the two species ($T$)[25], which we have estimated to be $2.89 \times 10^9$ (see "Methods"). We obtained $V_m$ based on the 16 MA lines with MA + WGS-based estimates of SNV rates and corrected the impact of deleting *MSH2* as in the above analysis. We found $D^2/V_m$ to be at least 4000 times smaller than $T$ (Table 1). We also respectively estimated $V_{mL}$ and $V_{mH}$ using the 16 MA lines with MA + WGS-based estimates of $\mu$. Again, we found $D^2/V_{mL}$ to be at least 400 times smaller and $D^2/V_{mH}$ at least 2400 times smaller than $T$ (Table 1), demonstrating selection against lowering as well as increasing $\mu$ in the divergence of *Saccharomyces* species. The above conclusion holds irrespective of whether the SNV rates are log$_{10}$-transformed (Table 1) or not (Supplementary Table 4).

**$\mu$ is well above the drift barrier in diverse organisms**. To directly confirm that $\mu$ is maintained above the drift barrier, we estimated the drift barriers for a diverse set of organisms including *S. cerevisiae* (see "Methods"). The drift barrier is commonly considered in terms of the number of mutations per functional genome per generation ($U$), which equals $\mu G$, where $G$ is the size of the functional genome or the number of nucleotides where mutations would be subject to selection. Although the drift barrier varies by the parameters assumed, our estimates were based on the best available information that was also used in the formulation of the BDH[18]. In every species examined, $U$ is substantially higher than the drift barrier, often by one to several orders of magnitude (Table 2). For example, *S. cerevisiae*'s $U$ is over 3000 times the estimated drift barrier.

Note that estimating the drift barrier requires knowing $N_e$, which is typically inferred from the synonymous nucleotide diversity under the assumption that synonymous mutations are

**Table 2 Observed SNV mutation rates per functional genome per generation and the corresponding drift barriers of model organisms.**

| Species | $N_e$ | Drift barrier | Observed mutation rate |
|---|---|---|---|
| *Escherichia coli* | $4 \times 10^8$ | $4 \times 10^{-7}$ | $8.3 \times 10^{-4}$ |
| *Bacillus subtilis* | $6 \times 10^7$ | $6 \times 10^{-6}$ | $1.2 \times 10^{-3}$ |
| *Saccharomyces cerevisiae* | $1 \times 10^7$ | $5 \times 10^{-7}$ | $1.7 \times 10^{-3}$ |
| *Schizosaccharomyces pombe* | $1 \times 10^7$ | $1 \times 10^{-6}$ | $1.6 \times 10^{-3}$ |
| *Chlamydomonas reinhardtii* | $4 \times 10^7$ | $3.8 \times 10^{-5}$ | $3.8 \times 10^{-2}$ |
| *Arabidopsis thaliana* | $4 \times 10^5$ | $3.8 \times 10^{-3}$ | $2.9 \times 10^{-1}$ |
| *Drosophila melanogaster* | $1 \times 10^6$ | $1.5 \times 10^{-3}$ | $1.9 \times 10^{-1}$ |
| *Mus musculus* | $1 \times 10^5$ | $1.5 \times 10^{-2}$ | $7.0 \times 10^{-1}$ |
| *Homo sapiens* | $1 \times 10^4$ | $1.5 \times 10^{-1}$ | 2.9 |

See "Methods" for references and drift barrier estimations.

neutral (see "Methods"). If synonymous mutations are overall slightly deleterious as has been suggested[26], $N_e$ would have been underestimated and drift barrier overestimated, rendering the true difference between the observed $U$ and the drift barrier even larger than that shown in Table 2. In other words, our conclusion based on Table 2 is conservative.

**Discovery of *PSP2* as a mutator gene**. Antimutator genes lower $\mu$, so studying them helps understand the molecular basis of high replication fidelity. About 30 antimutator genes are known in *S. cerevisiae*[27]. By contrast, mutator genes, whose normal functions are to increase $\mu$, have not been reported. Note the distinction between mutator genes and mutator alleles, the latter being loss-of-function alleles of antimutator genes. That $\mu$ is substantially lower in some MA lines than the progenitor suggests the existence of mutator genes that were crippled in MA; their discovery would help understand the mechanisms of mutagenesis and $\mu$ regulation. We originally identified candidate mutator genes by screening genes that were more frequently mutated in low-$\mu$ lines than in high-$\mu$ lines among MA lines with *CAN1*-based $\mu$ estimates (Supplementary Table 5), and picked four candidates (*RAD9*, *YFL013W-A*, *PSP2*, and *MSH4*) based on their ranks from the screening and annotated functions for a follow-up study. We subsequently found that some MA lines had erroneous *MSH2*, so reinserted *MSH2* followed by re-estimation of *CAN1*-based $\mu$. Based on these new estimates of $\mu$ (Fig. 2a), the four genes are ranked 1362, 911, 58, and 76, respectively. We respectively knocked out these four genes in the progenitor and measured the *CAN1*-based $\mu$. We found the *PSP2*-lacking strain to exhibit a significantly reduced $\mu$ ($P < 0.05$; Fig. 3a) and

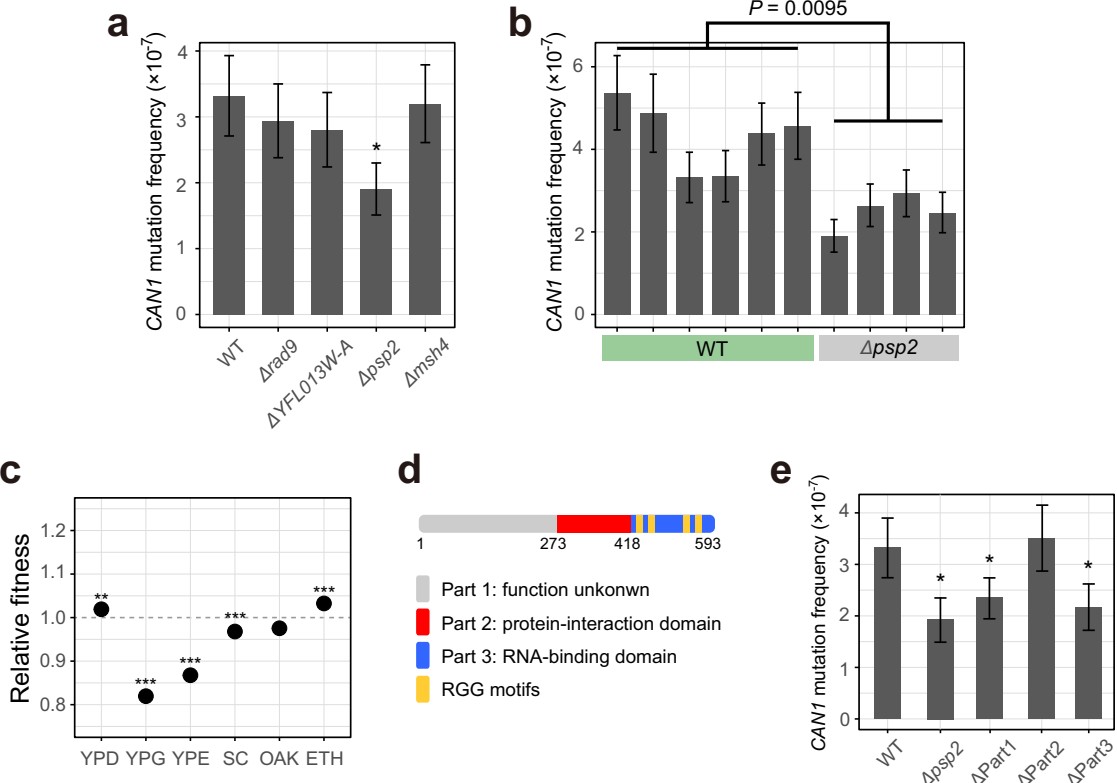

**Fig. 3 Discovery of *PSP2* as a mutator gene. a** *CAN1* mutation frequency of gene-deletion strains and the progenitor (WT) based on 72 biologically independent cell cultures from each strain. Bars represent mean estimates, error bars show 95% confidence intervals, and an asterisk indicates a significantly lower mutation frequency than that of the WT (*P* < 0.05 based on non-overlapping 95% confidence intervals). **b** *CAN1* mutation frequencies of six replicate measures in the progenitor (WT) and four replicate measures in the strain with *PSP2* knocked out. Bars represent mean estimates, error bars show 95% confidence intervals, and the *P* value is based on two-tailed Wilcoxon rank-sum test. **c** The fitness of the *PSP2*-deletion strain relative to that of the WT (dotted line) in six different environments. YPD, rich medium; YPG, glycerol medium; YPE, ethanol medium; SC, synthetic complete medium; OAK, synthetic oak exudate medium; ETH, rich medium with 6% ethanol[32]. **, *Q*-value < 0.01, ***, *Q*-value < 0.001 in the original genome-wide study of fitness effects of gene deletions[32]. **d** Functional domains of PSP2 based on the literature. Numbers indicate amino acid positions. **e** *CAN1* mutation frequency in *PSP2*-truncation strains based on 72 biologically independent cell cultures from each strain. Bars represent mean estimates and error bars show 95% confidence intervals. * indicates significantly lower than the progenitor (WT) (*P* < 0.05 based on non-overlapping 95% confidence intervals).

confirmed it by additional replications (*P* = 0.0095, Wilcoxon rank-sum test; Fig. 3b). That removing *PSP2* reduces *μ* by 42% (Fig. 3b) suggests that it is a major mutator gene. *PSP2* (polymerase suppressor 2) was originally discovered from a screening of rescuers of heat-sensitive mutations in *POL1* and *POL3*, which encode the catalytic subunit of DNA polymerase I and δ, respectively[28]. PSP2 is an RNA-binding protein and promotes P-body assembly[29,30]. Under nitrogen starvation, PSP2 binds to the mRNAs of *ATG1* and *ATG13* to promote their translation and autophagy; deleting *PSP2* reduces the synthesis of ATG1 and ATG13, autophagy activity, and cell survival[31]. Yeast grew faster under some conditions but slower under other conditions upon *PSP2* deletion[32] (Fig. 3c). Note that removing *PSP2* reduced *μ* in a medium similar to SC (synthetic complete) (Fig. 3b), where the knockout slowed yeast growth (Fig. 3c), but the causal relationship between the slowed growth and reduced *μ* is unclear.

PSP2 can be divided into three segments: the N-terminal segment has unknown functions, the middle segment interacts with translation initiation factors, and the C-terminal segment harbors four RGG motifs and binds to RNAs (Fig. 3d); the middle and C-terminal segments are required for PSP2's role in autophagy[31]. We respectively deleted from the progenitor the DNA sequences corresponding to the three segments of PSP2, followed by *CAN1*-based *μ* estimation. We found that the N-terminal and C-terminal segments but not the middle segment

are required for PSP2's activity in increasing *μ* (Fig. 3e), suggesting that PSP2 regulates *μ* through RNA binding but not protein interaction.

**Mutation spectrum has been shaped by selection**. To investigate the potential role of natural selection in shaping yeast's mutation spectrum, we compared the variance ($V_g$) in a component of the mutation spectrum among five divergent natural yeast strains having published MA + WGS data (Supplementary Fig. 3), with the corresponding mutational variance per generation estimated from the 16 MA lines with MA + WGS data. Because haploid and diploid progenitors show similar mutational spectrums (Fig. 4), we analyze the MA lines and natural strains regardless of their ploidy. Even under the most generous calculation, $V_g/V_m$ ($3.07 \times 10^4$) of the proportion of mutations that are SNVs is orders of magnitude smaller than the neutral expectation of $4 \times 10^7$ (Supplementary Table 6). In fact, the variance of the proportion of SNVs is smaller among the five natural strains than among the 16 MA lines (Fig. 4a), despite that the numbers of generations separating the natural strains are much greater than those separating the MA lines even after the correction for the increased mutagenesis of MA lines induced by deleting *MSH2*. Similar results were found regarding the proportion of insertions (maximal $V_g/V_m = 3.57 \times 10^3$) and that of deletions (maximal $V_g/V_m = 3.27 \times 10^4$) (Fig. 4a, Supplementary

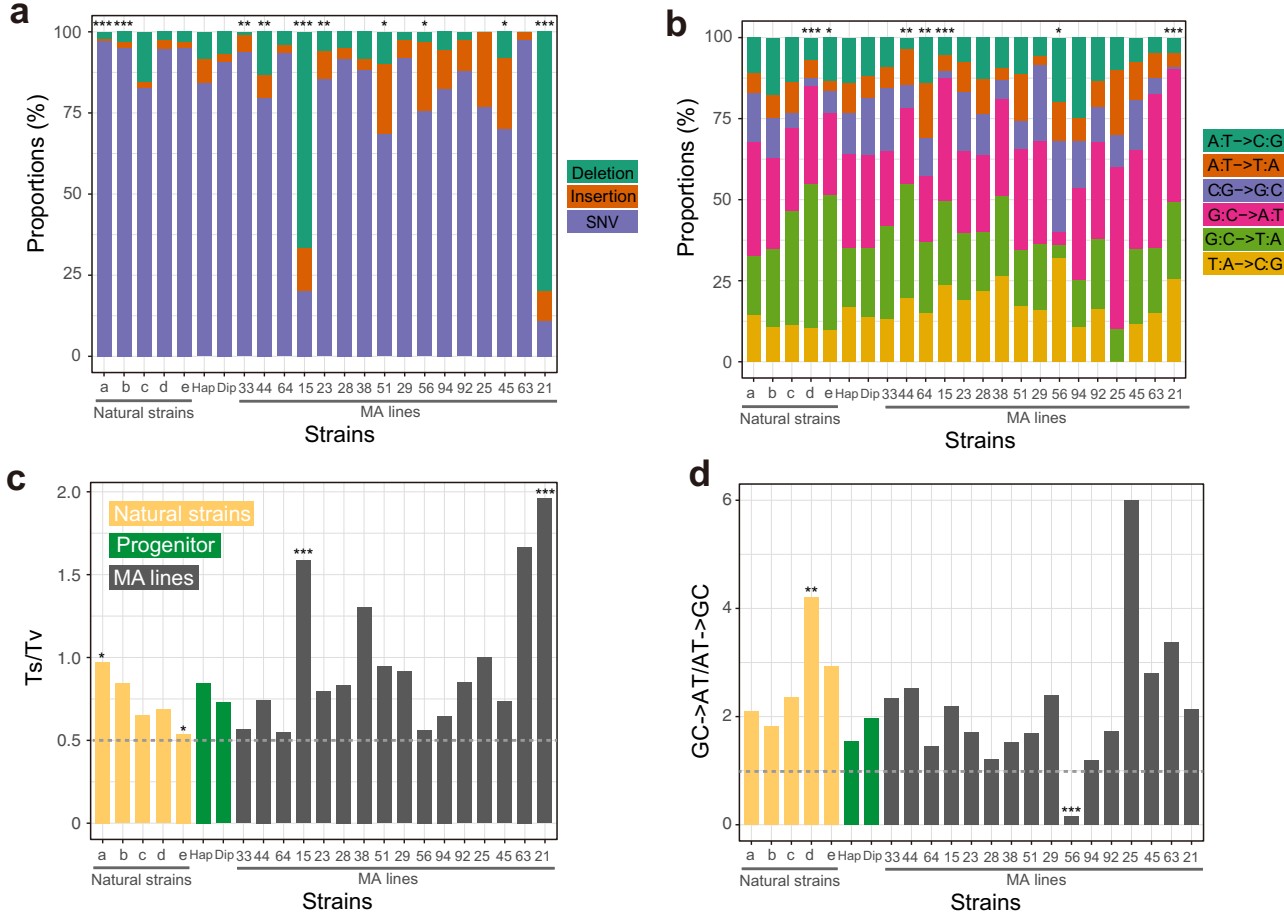

**Fig. 4 Molecular spectra of mutations in 16 MA lines and 5 natural yeast strains estimated by MA + WGS. a** Proportions of SNVs, insertions, and deletions among all mutations. **b** Relative proportions of the six types of SNVs. **c** Number of transition mutations relative to the number of transversion mutations (Ts/Tv). **d** Number of GC → AT mutations relative to the number of AT → GC mutations. In **c** and **d**, the gray dotted line shows the random expectation. In all panels, the small letters on the *X*-axis refer to natural strains (a DBY4974/DBY4975; b SEY6211; c SK1/BY; d DBVPG6765; and e YPS128/DBVPG6765), "Hap" refers to the progenitor in the haploid form, "Dip" refers to the progenitor in the diploid form, and the numbers refer to IDs of MA lines. Two-tailed chi-squared test is performed between each strain and the progenitor: *, $P < 0.05$; **, $P < 0.01$; ***, $P < 0.001$.

Table 6). Thus, the fractions of SNVs, insertions, and deletions among all mutations have been under stabilizing selection. Note, however, that because the three fractions must add to 1, the three fractions may not be subject to three separate stabilizing selections.

There are six different types of SNVs (Fig. 4b) and we found evidence for stabilizing selections on each of the six fractions (maximal $V_g/V_m$ ranging between $3.03 \times 10^4$ and $1.35 \times 10^6$; Supplementary Table 6). Again, there may not be six separate stabilizing selections because the six fractions must add up to 1.

Two mutational biases are of special interest because of their universal presence across the tree of life. The first is the transition/transversion (Ts/Tv) bias. Transitions are changes between purines or between pyrimidines, whereas transversions are changes between a purine and a pyrimidine. In almost all species examined, the mutational Ts/Tv ratio exceeds 0.5, the random expectation[33]. We found evidence for stabilizing selection of Ts/Tv (Fig. 4c); the maximal $V_g/V_m$ equals $9.13 \times 10^4$ (Supplementary Table 6). In particular, Ts/Tv is significantly higher (or lower) than that of the progenitor in two (or zero) MA lines (Fig. 4c). Hence, the stabilizing selection appears to have mainly kept the mutational Ts/Tv ratio low.

The second bias, known as the AT mutational bias, refers to the observation that GC → AT mutations outnumber AT → GC mutations. The universality of this bias across all species examined

has led to the belief that it arises from the chemical nature of DNA irrespective of variations in replication and repair mechanisms[34]. We found that, in one MA line, the ratio of the number of GC → AT mutations to the number of AT → GC mutations is significantly different from that in the progenitor, and is reversed from >1 to <1 (Fig. 4d). Clearly, the AT mutational bias is subject to genetic control and is not a chemical necessity. Furthermore, $V_g/V_m$ for the AT mutational bias is at least 120 times lower than the neutral expectation (Supplementary Table 6), indicating that the bias has been maintained by stabilizing selection. Stabilizing selection on mutation spectrum is evident regardless of whether we $log_{10}$-transform the original trait values (Supplementary Table 7) or not (Supplementary Table 6).

## Discussion

Consistent with the prediction of the DBH, a strong negative correlation between $U$ and $N_e$ was previously observed across diverse organisms[12], but several considerations suggest that the actual correlation is likely substantially weaker[35] (see "Methods"). Regardless, our finding in yeast that $\mu$ is selectively maintained well above the drift barrier refutes the DBH and suggests the presence of the first- and/or second-order selection for higher $\mu$ (Fig. 1a). Indeed, experiments in *Escherichia coli* found that genotypes with relatively high $\mu$ often outcompete those with relatively low $\mu$ in 100 generations of evolution despite their lack of difference in fitness prior to

the evolution[36]. Similarly, a *S. cerevisiae* mutator strain surpassed the wild-type in ~250 generations of evolution in large but not small co-cultures[37]. In Lenski's 50,000-generation *E. coli* experimental evolution in a low-glucose environment, 6 of 12 populations increased in $\mu$ and they adapted faster than the other 6 populations[38]. These observations support that modifiers raising $\mu$ can be fixed as a result of second-order selection. Furthermore, under certain conditions, the optimal $U$ resulting from the two opposing second-order selections (Fig. 1a) is predicted to decline with $N_e$ in asexuals (Supplementary Fig. 4, see "Methods"), which could partially explain the reported negative correlation between $U$ and $N_e$[12]. Nonetheless, the predicted optimal $U$ given $N_e$ appears lower than the corresponding observed $U$ (Supplementary Fig. 4). Furthermore, even in asexuals, the second-order selection for higher $\mu$ is episodic depending on the environment and the frequency of beneficial mutations, so $\mu$ likely fluctuates under its influence[7,8,11]. In sexuals, this selection is expected to be ineffective, because the mutation rate modifier becomes quickly unlinked with the beneficial mutation created and loses its selective advantage[6,9].

Experiments in several viruses have discovered a tradeoff between the speed and fidelity of genome replication[39,40], providing direct evidence for a cost of fidelity. We found that PSP2, a positive regulator of yeast autophagy, increases $\mu$ and that the autophagy-promoting and $\mu$-increasing activities both rely on PSP2's RNA-binding domain. While the mechanistic connection between these two activities is unclear, our finding suggests that the cost of fidelity could result from pleiotropy[41]; deleting PSP2 increases fidelity (Fig. 3b) but impairs autophagy and slows cell growth[31]. It is, however, unknown whether the fitness cost of fidelity is correlated with $N_e$ so could potentially create a negative correlation between $U$ and $N_e$, which seems possible at least in multicellulars. For example, $N_e$ is about ten times higher in mouse than in human (Table 2). There are fewer germ cell divisions per generation in mouse than in human, potentially rendering the demand and thus the cost of fidelity per cell division lower in mouse than in human; consistently, mutation rate per cell division is higher in mouse than in human[42]. Consequently, the fitness cost of fidelity per generation is also lower in mouse than in human, predicting a lower $\mu$ in mouse than in human (Fig. 1a), as is observed[42]. If the trend in human and mouse is generalizable, which seems plausible given the correlations between $N_e$ and many life-history traits, $\mu$ would generally decrease with $N_e$. Furthermore, $G$ is strongly negatively correlated with $N_e$ (see "Methods"). So, under the above model, $U$ would have a rather strong negative correlation with $N_e$. Hence, the DBH is not the only hypothesis that could explain a strong negative correlation between $U$ and $N_e$ if this correlation truly exists (see "Methods").

Our data do not allow us to distinguish between the first-order and second-order selections that lift $\mu$ from the drift barrier. Considering past theoretical results[7–9,11], we suggest that the first-order selection is more likely responsible for a $\mu$ that is stably above the drift barrier, while the second-order selection may further raise it episodically. While our experiments focused on yeast, the finding of a higher $U$ than the drift barrier across diverse organisms (Table 2) suggests that our conclusion is likely general. That the observed mutation rates of natural organisms are orders of magnitude above the theoretical minimums suggests the possibility of lowering their mutation rates through genome editing, which would have both theoretical and practical values.

Our data also provide evidence that the molecular spectrum of mutation has been selectively shaped, but it is unknown whether the selections directly or indirectly act on the spectrum and what the selective agents are. Further studies are needed to answer these questions. Mutation and selection are generally considered distinct evolutionary forces. Our finding that both the rate and spectrum of mutation are determined by actions of natural

selection somewhat blurs the separation of mutation from selection, which may offer new insights into evolution.

## Methods

**Strains and genetic manipulations**. We knocked out the *MSH2* gene from the haploid BY4741 strain of *S. cerevisiae*, referred to as the progenitor, by homologous recombination with KanMX, followed by selection on YPD (1% yeast extract, 2% peptone, and 2% dextrose) plates with 0.5 g/L G418. The knockout of *MSH2* was confirmed by Sanger sequencing. This *MSH2*-lacking strain was used to initiate the first round of MA. Upon the completion of the MA, *MSH2* was inserted back to the resultant strains using CRISPR-Cas9 genome editing[43]. Specifically, the wild-type *MSH2* from BY4741 was used as the repair fragment, and three different guide RNAs targeting KanMX were used. Transformation was performed in each MA line for up to three times and was confirmed by Sanger sequencing of the reinserted locus. Restoration of *MSH2* was successful in only 60 of the 93 MA lines, probably due to reduced transformation efficiencies in the MA lines.

Four candidate mutator genes (*RAD9*, *YFL013W-A*, *PSP2*, and *MSH4*) and three segments of *PSP2* were also respectively removed from the progenitor (with intact *MSH2*) using CRISPR-Cas9. The start (or stop) codon was left unchanged when we removed the DNA of the N-terminal (or C-terminal) segment of PSP2. Primers used in this study can be found in Supplementary Table 8.

**MA and whole-genome sequencing**. The strategy of two rounds of MA was previously used in animals to probe the change of $\mu$ as a result of MA[44,45]. In the first round of MA in our study, 96 parallel lines were established from the BY4741 without *MSH2*. Cells were propagated at 30 °C on YPD plates. A single-cell bottleneck was applied to each line every 48 h, where a randomly picked average-size colony was streaked onto a new plate. Each line went through a total of 80 bottlenecks, and 93 of the 96 lines survived in the end. The total number of generations each MA line went through was estimated by the number of generations between bottlenecks multiplied by the number of bottlenecks. We estimated the number of generations between bottlenecks by counting the number of cells in an average-size colony and assuming exponential growth, and took the average of the estimates prior to and after MA. The $N_e$ of an MA line equals the harmonic mean of the number of cells per generation. In our experiment, the average between-bottleneck number of generations is 21, so $N_e$ equals $21/(1/1 + 1/2 + 1/2^2 + \ldots + 1/2^{21})$ ≈10. The genomes of the 93 MA lines and the progenitor in the *MSH2*-lacking background were sequenced.

A total of 18 strains, including 16 of the above 93 MA lines, BY4741 (haploid), and BY4743 (diploid), all with intact *MSH2*, were subject to the second round of MA. Four to 20 replicate lines were established for each strain. Cells were propagated at 30 °C on SC (synthetic complete) plates, similar to that used in the fluctuation test. The total time in the second round of MA for all lines was kept at ~100 days and the number of generations between bottlenecks was kept at ~20. The between-bottleneck duration was different among the 18 strains because of their different generation times. It was 48 h in BY4741, BY4743, MA28, and MA38, 72 h in MA15, MA21, MA23, MA25, MA29, MA33, MA44, MA63 and MA92, and 96 h in MA45, MA51, MA56, MA64, and MA94. The genomes of 209 MA lines at the end of the second round of MA and their 18 ancestral strains were sequenced. The number of generations each MA line went through was estimated in the same way as in the first round of MA.

For each sample to be sequenced, the genomic DNA was extracted using MasterPure Yeast DNA Purification Kit (Lucigen; Cat. No. MPY80200). Library was constructed using Nextera DNA Flex Library Prep kit (Illumina; Cat. No. 20018705). Paired-end reads (2 × 150 bases) were generated on Illumina Hiseq 4000 platform by Admera Health (www.admerahealth.com).

**Identification of mutations and verification by Sanger sequencing**. Sequencing reads from each sample were first mapped to the *S. cerevisiae* reference genome (version R64-2-1) by Burrows-Wheeler Aligner[46]. Duplicate marking and local realignment around indels were carried out using Genome Analysis Toolkit (GATK)[47]. SNVs and indels shorter than 50 nucleotides were called by GATK HaplotypeCaller. Variants that differ between each MA line and its ancestral strain were retained when they met the following criteria: (i) a variant must be homogeneous because the MA lines in this study were haploid, (ii) a variant site must be covered by at least five reads in both the MA line and the ancestor, (iii) a variant must be supported by both forward and reverse reads, and (iv) a variant must have a quality score no lower than 50. Mutation rate was computed by (number of mutations in a sample)/(number of callable sites)/(number of generations in MA), where callable sites were defined as genomic sites covered by at least five reads.

Twenty shared mutations between MA lines identified in the first round of MA were randomly chosen for verification by Sanger sequencing. For each mutation, Sanger sequencing was performed in both the sample with the mutation and its ancestor, and the mutation was considered confirmed by Sanger sequencing if both results agreed with the results from Illumina sequencing. Polymerase chain reaction and Sanger sequencing were successful in 16 of the 20 cases, and 15 of the 16 mutations were confirmed by Sanger sequencing.

The impact of deleting *MSH2* on mutation rates were assessed by comparing the mutations accumulated in the first round of MA in the *MSH2*-lacking background with those in the second round of MA of the progenitor with intact *MSH2*. Deleting *MSH2* increased the SNV rate per site per generation by 16 times,

indel rate per site per generation by 580 times, and the total rate of SNVs and indels by 104 times.

**Confirmation of the infrequency of selection in the first round of MA**. Because the $N_e$ of the MA lines was about 10, while most mutations are expected to have a fitness effect on the order of 1% or smaller[21], selection should be infrequent during the MA. These infrequent selections likely concentrated in the one-sixth of yeast genes known as essential genes, because loss-of-function mutations in essential genes cannot accumulate. To confirm the infrequency of selection, we compared the genomic distributions of the observed mutations in the 93 MA lines with the corresponding random expectations. The fraction of SNVs located in genic regions is 73%, slightly but significantly below the random expectation of 74% ($P = 0.0046$, binomial test). The fraction of coding SNVs that are nonsynonymous is 70%, also slightly but significantly below the random expectation of 76% ($P < 0.001$, binomial test). While 16% of all homonucleotide runs reside in genic regions, a slightly lower fraction of indels (14%) occurred in genic regions ($P = 0.002$, chi-squared test). Together, these results confirm that selection was present but infrequent in our MA experiment. The infrequent selection may cause a slight underestimation of $V_m$ (under both the DBH and conventional model), rendering our inference of stabilizing selection more conservative.

**Fluctuation test**. *CAN1*-based fluctuation test was performed following Lang[48] with a few modifications. *CAN1* encodes an arginine transporter; cells must carry loss-of-function mutations in *CAN1* to be able to grow in the presence of canavanine, a toxic arginine analog. The strain being tested was precultured in SC-Arg liquid medium for 48 h. Cells were then diluted and transferred to a 96-well plate with an initial cell number of ~1000 per well. Each well in the 96-well plate contained 100 μl fresh SC-Arg medium. The plate was sealed with an aluminum film and incubated at 30 °C with shaking for 72 h. Then, 72 100-μl cultures were spot-plated onto the selection plates (SC-Arg with 60 μg/ml canavanine), while the remaining 24 100-μl cultures were pooled followed by cell counting by a hemocytometer. About 1000 cells from this pool were plated onto a SC-Arg plate (without canavanine) to test plating efficiency. The canavanine plates with cell cultures were first dried in a sterile hood, followed by incubation at 30 °C for 72 h. Finally, the mutant colonies on each plate were manually counted. Because the growth rates of the MA lines were low, the incubation time in this step was longer than the usually used 49 h. We subjected all 60 MA lines with reinserted *MSH2* to the fluctuation test, but only 49 of them grew in the medium. We also subjected the progenitor (before the deletion of *MSH2*) to the fluctuation test.

*CAN1*-based $\mu$ was estimated by bz-rates[49], a web tool that uses an empirical probability generating function to estimate the number of mutations per culture[50] with correction for plating efficiency. The mutation frequency presented is the probability of loss-of-function mutation in *CAN1* per cell division. No *CAN1* mutant was observed in two MA lines, and their $\mu$ values were calculated by assuming the observation of one mutant colony to allow plotting $\mu$ in a logarithmic scale. The same practice was employed in estimating $V_m$, which rendered our selection tests more conservative. The number of mutants per culture in the fluctuation test follows the Luria–Delbrück distribution, which is a highly skewed, non-normal distribution[50]. To statistically evaluate the difference in $\mu$ between two strains, we used the 95% confidence intervals (CI) of $\mu$ estimated by bz-rates[49]; two strains with non-overlapping 95% CIs were regarded as having significantly different $\mu$.

**$N_e$ of S. cerevisiae**. The SNV rate of *S. cerevisiae* was estimated from MA lines to be $\mu = 1.95 \times 10^{-10}$ per site per generation in YPD[15]. A species-wide population genomic survey of *S. cerevisiae*[51] found that the nucleotide diversity per site ($\pi$) is substantially lower at nonsynonymous (0.0014), intronic (0.0027), and intergenic (0.0037) sites than at synonymous sites (0.0091). Under the assumption that synonymous mutations are neutral, $N_e$ was estimated by $\pi_S/(4\mu) = 1.17 \times 10^7$. It is possible that $\pi_S$ is smaller than the neutral nucleotide diversity because of selection at synonymous sites, which renders our estimate of $N_e$ smaller than its true value and our inference of stabilizing selections of mutation rates and spectrum conservative.

**Estimation of $V_m$, $V_g$, and $D^2$**. $V_m$ of a phenotypic trait such as $\mu$ is the variance of $\mu$ among MA lines per generation. $V_{mL}$ (or $V_{mH}$) is the corresponding variance calculated using only MA lines with lower (or higher) $\mu$ than that of the progenitor. Because $\mu$ is higher in the *MSH2*-lacking MA lines than in natural strains, we employed three corrections by respectively assuming that deleting *MSH2* caused the same fold change in the rate of each mutation type as the observed fold change of the total rate of SNVs and indels ($V_{m1}$), as that of indels ($V_{m2}$), and as that of SNVs ($V_{m3}$). Specifically, the corrected numbers of generations became 119,850 in estimating $V_{m1}$, 667,636 in estimating $V_{m2}$, and 18,578 in estimating $V_{m3}$, respectively. In the top half of Table 1, $V_m$, $V_{mL}$, and $V_{mH}$ were estimated using the *CAN1*-based $\mu$ of MA lines from the present study, while $V_g$ was estimated using published *CAN1*-based $\mu$ of seven diverse natural strains of *S. cerevisiae*[14] (Supplementary Table 3, Supplementary Fig. 3). In the bottom half of Table 1, $V_m$, $V_{mL}$, and $V_{mH}$ were estimated using the MA + WGS-based SNV rates of MA lines from the present study, while $D^2$ was the squared difference in MA + WGS-based SNV mutation rate between *S. cerevisiae*[15] and *S. paradoxus*[24]. In the above analysis, we either log₁₀-transformed $\mu$ before computing $V_m$, $V_g$, and $D^2$ (Table 1) or used the original $\mu$ values without transformation (Supplementary Table 4).

In Supplementary Table 6, $V_m$ was estimated using the MA + WGS-based mutation rates of MA lines from the present study, while $V_g$ was estimated using published MA + WGS-based mutation rates of five diverse natural strains of *S. cerevisiae*[19,24,52,53] (Supplementary Fig. 3). We presented results from both log₁₀-transformed values (Supplementary Table 7) and untransformed values (Supplementary Table 6).

To test if $V_g/V_m$ is significantly smaller than the neutral expectation, we bootstrapped MA lines as well as natural strains 10,000 times; $P$-value is the fraction of times when $V_g/V_m$ computed from a bootstrap sample exceeds the neutral expectation. To test if $D^2/V_m$ is significantly smaller than the neutral expectation, we bootstrapped MA lines 10,000 times; $P$ value is the fraction of times when $D^2/V_m$ computed from a bootstrap sample exceeds the neutral expectation. The $V_g$ and $V_m$ calculated here are phenotypic variances, including the genetic component and estimation error, because the environment is fixed. The phenotypic variance caused by estimation error should be similar for natural strains and MA lines because of the use of the same phenotyping method. Because the phenotypic variance is greater for MA lines than for natural strains, the fraction of phenotypic variance contributed by genetics is greater for MA lines than for natural strains. Hence, $V_g/V_m$ is overestimated when computed using phenotypic variance instead of genetic variance, which renders our conclusion that $V_g/V_m$ is smaller than the neutral expectation conservative.

**Number of generations separating S. cerevisiae and S. paradoxus**. The number of generations separating *S. cerevisiae* and *S. paradoxus* was estimated by dividing the nucleotide sequence divergence between the two species per synonymous site ($d_S$) by the mean SNV mutation rate of *S. cerevisiae* and *S. paradoxus*. $d_S$ has been estimated to be 0.3868[54] and the reported SNV mutation rates in these two species are $1.95 \times 10^{-10}$ and $7.27 \times 10^{-11}$ per site per generation, respectively[15,24]. Hence, the number of generations separating the two species is $2.89 \times 10^9$. As noted above, $d_S$ may underestimate the neutral divergence between the two species, which renders our stabilizing selection conclusion conservative.

**The mutation rate drift barriers of various species**. For (haploid) asexuals, the drift barrier ($U_0$) is reached when the mutation rate reduction per functional genome per generation by a modifier ($\Delta U = \lambda U_0$) equals $1/N_e$, where $\lambda$ is the fractional reduction of the mutation rate and is assumed to be 0.1[18]. Hence, the drift barrier is $U_0 = 10/N_e$. For diploid asexuals, because $\Delta U = 2\lambda U_0$, $U_0 = 5/N_e$. Let $\Delta m$ be the per generation rate of mutational production of modifiers decreasing the mutation rate minus the corresponding rate of production of modifiers increasing the mutation rate. When $U$ approaches $U_0$, $\Delta m$ is likely negative, which will cause an increase in $U_0$[18]. However, because the magnitude of $\Delta m$ is expected to be much smaller than $U_0$[18], the effect of $\Delta m$ on $U_0$ is negligible.

For diploid sexuals at the drift barrier, $\Delta U = 2\lambda U_0 = 1/(2N_e s)$, where $s$ is the mean selective disadvantage of deleterious mutations in the heterozygous state[18] and is assumed to be 0.01[21]. So, $U_0 = 1/(4N_e s\lambda) = 250/N_e$. In sexuals, the effect of $\Delta m$ on $U_0$ is amplified by $1/s$ times, so it is possible that $U_0$ is increased to some extent due to a negative $\Delta m$. However, it is extremely unlikely that the absolute value of $\Delta m/s = 100\Delta m$ exceeds $5U_0$, because it is difficult to imagine that more than 5% of mutations create mutation rate modifiers. Therefore, $U_0$ should not exceed 6 times the above estimate, or $1500/N_e$.

We obtained the $N_e$ estimates of *E. coli*[55], *Bacillus subtilis*[12], *Schizosaccharomyces pombe*[56], *Chlamydomonas reinhardtii*[57], *Arabidopsis thaliana*[58], *Drosophila melanogaster*[59], *Mus musculus*[60], and *Homo sapiens*[61] from the literature. The $N_e$ of *S. cerevisiae* was assumed to be $10^7$, as estimated above. *S. cerevisiae* is considered a diploid asexual because it is normally a diploid but undergoes rare sexual reproduction (once per 1000 generations) in the wild[62]. *S. pombe* is considered a haploid asexual because it is normally a haploid and very rarely undergoes sexual reproduction (once per 0.6 million generations) in the wild[62]. *Arabidopsis thaliana* has become largely selfing since about 1 million years ago[63], but we here still consider it outcrossing so that our argument for a $U$ that is higher than the drift barrier is more conservative. *Chlamydomonas reinhardtii* is normally a haploid, but how often it undergoes sexual reproduction in nature is unknown[64]. To be conservative in our argument, we considered it a sexual haploid.

The observed $U$ is measured by the number of SNV mutations per functional genome per generation. The functional genome size $G$ is the number of nucleotides in the genome that are subject to natural selection. To be conservative, $G$ was assumed to equal the total coding sequence length except for *D. melanogaster*, *M. musculus*, and *H. sapiens*. For these three species, $G$ was estimated by the total number of nucleotides in autosomes multiplied by the faction of sites under purifying selection, which was previously estimated to be 48.7% for *D. melanogaster*[65], 7.2% for *M. musculus*[65], and 8.2% for *H. sapiens*[66], respectively. We obtained the estimates of SNV mutation rate per site per generation of *E. coli*[67], *B. subtilis*[68], *S. cerevisiae*[15], *S. pombe*[69], *C. reinhardtii*[70], *A. thaliana*[71], *D. melanogaster*[72], *M. musculus*[42], and *H. sapiens*[42] from the literature.

**On the negative correlation between U and $N_e$**. For three reasons, the cause of the reported[12] negative correlation between $U$ and $N_e$ is uncertain. First, $N_e$ is typically estimated by dividing the neutral nucleotide diversity by $k\mu$, where $k$ is 4 for diploids and 2 for haploids, and the neutral nucleotide diversity is typically approximated by the synonymous nucleotide diversity $\pi_S$. Hence, any estimation

error of $\mu$ influences the estimates of $U$ and $N_e$ in opposite directions, creating a spurious negative correlation between them[35]. Second, synonymous mutations are not immune to selection[26]. Due to the abundance of negative selection and the increase of selection intensity with $N_e$, it is possible that the neutral nucleotide diversity is underestimated by $\pi_S$ and that the extent of the underestimation rises with $N_e$. In other words, the larger the $N_e$, the greater the underestimation of $N_e$, which increases the apparent effect of $N_e$ on $U$. Finally, the negative correlation between $U$ and $N_e$ is partially due to the negative correlation between $G$ and $N_e$. For instance, $\log_{10} N_e$ has a linear correlation coefficient of $-0.61$ ($P = 4.0 \times 10^{-4}$) with $\log_{10}(\text{proteome size})$ and $-0.77$ ($P = 1.1 \times 10^{-6}$) with $\log_{10}(\text{genome size})$ when analyzed using published data[12].

**Optimal $U$ under the two opposing second-order selections in asexuals**. Raising $U$ increases the number of beneficial mutations as well as that of deleterious mutations. Orr found that, in asexuals, the optimal $U$, which is the $U$ value corresponding to the highest speed of adaptation, is the harmonic mean of the coefficient of selection ($s > 0$) against deleterious mutations[7]. Because deleterious mutations with $s$ smaller than $1/N_e$ are effectively neutral, the range of $s$ for mutations that are selected against enlarges with $N_e$, which causes a decrease of the harmonic mean of $s$ and hence the optimal $U$ with $N_e$. To show this trend numerically, we sampled 100,000 $s$ values from a gamma distribution with mean equal to 0.01 and shape parameter $\alpha = 0.1, 0.2$, or $0.5$. The harmonic mean of $s$ among all sampled $s$ values that are larger than $1/N_e$ was computed. This harmonic mean is the optimal $U$. We considered various $N_e$ values and various $\alpha$ values.

**Reporting summary**. Further information on research design is available in the Nature Research Reporting Summary linked to this article.

## Data availability

The sequencing reads generated have been deposited to NCBI SRA under the accession number PRJNA735524. All other data are presented in the paper and associated supplementary materials. Source data are provided with this paper.

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

## Acknowledgements

We thank Liangke Gou and Leonid Kruglyak for providing the raw replicate mutation rate estimates of seven natural strains of yeast and Alex Kondrashov, Wenfeng Qian, and members of the Zhang laboratory for valuable comments. This work was supported by the U.S. National Institutes of Health research grant R35GM139484 to J.Z.

## Author contributions

H.L. and J.Z. designed the study and wrote the paper. H.L. performed the research and analyzed the data.

## Competing interests

The authors declare no competing interests.
