## [Peer Review File · Nature Communications]

REVIEWERS' COMMENTS

Reviewer #1 (Remarks to the Author):

As basically always the case I find manuscripts coming from the Zhang laboratory fascinating and thought-provoking. The claims are exciting and the experiments are super cleve. This one is no exception.

The experimental design is quite brilliant. First do MA in a mutator genotype -> get a large number of mutations -> fix the mutator genotype -> carry out fluctuations tests to measure mutation rate and WGS validation of the mutation rates per strain. Fascinatingly, it turns out to be very easy to generate yeast strains that have lower mutation rate and shift the mutation profile. This result alone makes the paper worth publishing.

Despite all of this and despite the fact that I personally do not believe in the drift barrier hypothesis, I found the argument that this result in and of itself disproves the drift-barrier hypothesis unconvincing. In order to show this you need to know exactly where the barrier is and I for one do not know any convincing measurements of the barrier. Ne based on π is not a great measure especially for yeast which violates all assumptions of such a calculation strongly.

Because of these reasons I don't recommend the publication of this ms unless the claims are weakened substantially. Congratulations on the beautiful results and experimental design though. Very impressive.

Reviewer #2 (Remarks to the Author):

In this study, Liu and Zhang tested the drift–barrier hypothesis using mutation accumulation experiments. They found that the mutation rate could be substantially reduced under random mutation accumulation, indicating that the apparent mutation rate has not been restricted by the drift–barrier. They further identified a mutation gene, *PSP2*, upon the deletion of which the mutation rate was significantly reduced. They also offered an alternative explanation for the well-known correlation between the effective population size and the mutation rate—the trade-off between replication fidelity and speed. In addition to the mutation rate, they also showed that the molecular spectrum of mutation was under stabilizing selection.

In summary, the authors provided extensive experimental evidence, together with theoretical analyses, making the study solid and convincing. The question to be addressed here is fundamental to almost every aspect of biology. The paper is well written, and the distinctions between first- vs. second-order selection, between sexual vs. asexual reproduction were fairly introduced and discussed. I would be happy to see it published in Nature Communications as soon as possible.

Reviewer #3 (Remarks to the Author):

The authors report an experimental test of a fundamental hypothesis in evolutionary biology, which is that the mutation rate should evolve to the point where direct selection to reduce the mutation rate so as to reduce the input of deleterious mutations into the population is counterbalanced by the cumulative effects of random genetic drift and a weak upward mutational bias – the "Drift Barrier". The authors employ a beautiful "second-order mutation accumulation" strategy, taking full advantage of the power of the yeast system. They first constructed a set of 96 mutation accumulation (MA) lines in which they knocked out the MSH2 gene, thereby increasing the mutation rate (especially the indel rate). Then, they engineered an intact wild-type MSH2 allele into 66 of the surviving 93 MA lines using CRISPR, and measured the mutation rate of 48 of the reconstituted wild-type MA lines and of their (wild-type) common ancestor by means of classical Fluctuation Tests using a canavanine reporter system (CAN1). The results are admirably clear – 12 of the MA lines evolved significantly LOWER mutation rates than the ancestor (28 evolved significantly higher rates), and moreover, the lines that evolved lower rates in 11/12 cases had rates that were at least 40% lower than the ancestral rate, a result that is inconsistent with the predictions of the drift-barrier hypothesis (DBH). Next, (as if that wasn't cool enough!) they initiated a second set of MA lines derived from 13 of the 48 reconstituted wild-type MA lines, each line replicated into a set of 18-20 "second order" MA lines. These lines were then sequenced, and the genome-wide mutation rate and spectrum estimated. They observed a significant positive correlation between the CAN1 mutation rate and the SNV mutation rate, but not with the indel rate. Now here's where it gets really good. The 48 estimates of CAN1 mutation rate provide an unprecedentedly robust estimate of the mutational variance (VM) for mutation rate (with approximate lower and upper bounds set by the SNV rate and the indel rate, respectively). Then, they estimate the standing genetic variance (VG) in CAN1 mutation rate from a set of seven natural isolates. If mutation rate is a neutral trait, at mutation-drift equilibrium $VG=4NEVM$. The estimated VG for mutation rate is somewhere between 100X and 1000X below the predicted neutral rate. Further, since the mutation rate evolved both downward and upward, selection on mutation rate is evidently stabilizing and not uniformly directional (downward), as predicted by the drift barrier (but see below). In contrast, stabilizing selection is predicted by the leading (really the only) competing hypothesis of mutation rate evolution, the "cost of fidelity".

Next, they extended that analysis to the between-species comparison with the sister species *S. paradoxus*. The squared divergence between species in mutation rate $D2=VM*T$, where T is the time of divergence. Estimates of *S. paradoxus* mutation rate and the divergence time are available, and the upshot is that the difference in mutation rate between the two species is 100-1000 times less than expected from neutral divergence – again, the signature of stabilizing selection.

The authors then draw on data about mutation rate, "effective genome size" and N_e from a diversity of species, using the same assumptions as Lynch (2011) used in his initial analysis, to reach the conclusion that the mutation rate is usually well above the drift barrier.

But wait, there's more! The authors then fish out four candidate genes, do some clever functional genomics, and identify one – PSP2 – as a "mutator" gene, defined here as a gene whose NORMAL function is to INCREASE the mutation rate.

OK, so the bottom line: I love this work. It is clever, elegant, addresses a question of fundamental importance in evolutionary biology, and provides a clear answer to that question. My only caveat with

respect to the experimental work is this: I am not a microbiologist, and I am not competent to assess how rigorously generation time was estimated. Obviously, comparative studies of rates depend critically on what is in the denominator, i.e., the number of generations. I trust that one of the other reviewers IS competent to assess the rigor of the generation-time estimates, and that s/he will sign off.

Now, a thought with respect to the interpretation. The majority of the second-order MA lines DID evolve a higher mutation rate, so on average, the mutation rate did increase with MA, i.e., under relaxed selection. As predicted by the DBH. Now, I do buy the stabilizing selection argument, but it seems to me that the fitness function is not symmetric around the current mean (or in other words, there is a mutational bias), which I interpret to mean that there is both a directional (linear) and quadratic (stabilizing) component to selection on mutation rate. Now of course the drift barrier may have nothing to do with this, but I would argue that the evidence suggests that the long-term outcome of evolution under relaxed selection would be an increased average mutation rate, and where it equilibrates would seem to depend on the effectiveness of selection. My own sentiment is that the cost of fidelity HAS to be true (sort of like synergistic epistasis HAS to be true!), and I think the results of this study strengthen the case. But still: relax selection, and mutation rate drifts upward, on average. I think the authors need to explicitly acknowledge that fact.

And, I will note that this work has two historical antecedents. Way back in 2006, Ávila et al. (Genetics) did a second-order MA experiment in *Drosophila*, in which they reported that the mutation rate increased in the second-order MA lines (U estimated from fitness assays; I think their results are also consistent with synergistic epistasis, although they did not seem to think so). And in 2019, Saxena et al. (MBE) reported a second-order MA experiment with *C. elegans*, in which they observed that, on average, mutation rate increased under MA conditions, which they argued was consistent with the DBH. They also reported mutational heritability for mutation rate on the order of $10^{-3}/\text{gen}$ (just like for every other trait under the sun).

A few minor comments, noted by Page #/Line #.

1. P6/L145. The term "phenotypic variance" is a bit confusing in this context. I know what you mean, which is the genetic variance for a phenotypic trait, but "phenotypic variance" usually connotes VP, not VG. Maybe think about how to rephrase.
2. P8/L206. Your definition of "mutator" is different from usual. Again, the meaning is clear in context, but "mutator" alleles are typically considered to be alleles (presumably defective loss-of-function alleles) at what you call antimutator GENES that increase the mutation rate. This non-standard usage may lead to confusion down the road; an additional explanatory clause following the "30 antimutator genes" with respect to mutator alleles may smooth the path. Something to consider.

Response to reviewers

We thank the three reviewers for their valuable comments, which have helped improve our manuscript. Below please find our point-to-point response in blue.

Reviewer #1

As basically always the case I find manuscripts coming from the Zhang laboratory fascinating and thought-provoking. The claims are exciting and the experiments are super cleve. This one is no exception.

The experimental design is quite brilliant. First do MA in a mutator genotype -> get a large number of mutations -> fix the mutator genotype -> carry out fluctuations tests to measure mutation rate and WGS validation of the mutation rates per strain. Fascinatingly, it turns out to be very easy to generate yeast strains that have lower mutation rate and shift the mutation profile. This result alone makes the paper worth publishing.

Despite all of this and despite the fact that I personally do not believe in the drift barrier hypothesis, I found the argument that this result in and of itself disproves the drift-barrier hypothesis unconvincing. In order to show this you need to know exactly where the barrier is and I for one do not know any convincing measurements of the barrier. Ne based on π is not a great measure especially for yeast which violates all assumptions of such a calculation strongly.

Because of these reasons I don't recommend the publication of this ms unless the claims are weakened substantially. Congratulations on the beautiful results and experimental design though. Very impressive.

We thank the reviewer for the largely positive comments. Currently, the N_e of yeast is estimated by π at synonymous sites divided by 4μ , where π is nucleotide diversity per site and μ is mutation rate per site per generation. The assumption in this estimation is that synonymous sites are selectively neutral. It is likely that synonymous sites are overall under weak purifying selection, which means that N_e of yeast has been underestimated. Consequently, the real drift barrier would be lower than the current estimate. Thus, the true difference between the observed mutation rate and that predicted by the drift-barrier hypothesis would be even greater than that shown in our Table 2, meaning that our rejection of the drift-barrier hypothesis is even stronger than presented. Besides the comparison between the observed mutation rate and that predicted by the drift-barrier hypothesis, we provided two additional lines of evidence against the drift-barrier hypothesis. First, a substantial fraction of MA lines showed significantly reduced mutation rates after the removal of natural selection. Second, by separating V_m into V_{mL} and V_{mH} , we demonstrated selection for reducing the mutation rate as well as selection for elevating the mutation rate. Together, the above findings unambiguously refute the drift-barrier hypothesis. We have added the following paragraph on page 8 to clarify the issue.

“Note that estimating the drift barrier requires knowing N_e , which is typically inferred from the synonymous nucleotide diversity under the assumption that synonymous mutations are neutral (see Methods). If synonymous mutations are overall slightly deleterious as has been suggested²⁶, N_e would have been underestimated and drift barrier overestimated, rendering the true difference between the observed U and the drift barrier even larger than that shown in Table 2. In other words, our conclusion based on Table 2 is conservative.”

Reviewer #2

In this study, Liu and Zhang tested the drift–barrier hypothesis using mutation accumulation experiments. They found that the mutation rate could be substantially reduced under random mutation accumulation, indicating that the apparent mutation rate has not been restricted by the drift–barrier. They further identified a mutation gene, PSP2, upon the deletion of which the mutation rate was significantly reduced. They also offered an alternative explanation for the well-known correlation between the effective population size and the mutation rate—the trade-off between replication fidelity and speed. In addition to the mutation rate, they also showed that the molecular spectrum of mutation was under stabilizing selection.

In summary, the authors provided extensive experimental evidence, together with theoretical analyses, making the study solid and convincing. The question to be addressed here is fundamental to almost every aspect of biology. The paper is well written, and the distinctions between first- vs. second-order selection, between sexual vs. asexual reproduction were fairly introduced and discussed. I would be happy to see it published in Nature Communications as soon as possible.

We thank the reviewer for these positive comments. No response to this reviewer is required.

Reviewer #3

The authors report an experimental test of a fundamental hypothesis in evolutionary biology, which is that the mutation rate should evolve to the point where direct selection to reduce the mutation rate so as to reduce the input of deleterious mutations into the population is counterbalanced by the cumulative effects of random genetic drift and a weak upward mutational bias – the "Drift Barrier". The authors employ a beautiful "second-order mutation accumulation" strategy, taking full advantage of the power of the yeast system. They first constructed a set of 96 mutation accumulation (MA) lines in which they knocked out the MSH2 gene, thereby increasing the mutation rate (especially the indel rate). Then, they engineered an intact wild-type MSH2 allele into 66 of the surviving 93 MA lines using CRISPR, and measured the mutation rate of 48 of the reconstituted wild-type MA lines and of their (wild-type) common ancestor by means of classical Fluctuation Tests using a canavanine reporter system (CAN1). The results are admirably clear – 12 of the MA lines evolved significantly LOWER mutation rates than the ancestor (28 evolved significantly higher rates), and moreover, the lines that evolved lower rates

in 11/12 cases had rates that were at least 40% lower than the ancestral rate, a result that is inconsistent with the predictions of the drift-barrier hypothesis (DBH). Next, (as if that wasn't cool enough!) they initiated a second set of MA lines derived from 13 of the 48 reconstituted wild-type MA lines, each line replicated into a set of 18-20 "second order" MA lines. These lines were then sequenced, and the genome-wide mutation rate and spectrum estimated. They observed a significant positive correlation between the CAN1 mutation rate and the SNV mutation rate, but not with the indel rate.

Now here's where it gets really good. The 48 estimates of CAN1 mutation rate provide an unprecedentedly robust estimate of the mutational variance (VM) for mutation rate (with approximate lower and upper bounds set by the SNV rate and the indel rate, respectively). Then, they estimate the standing genetic variance (VG) in CAN1 mutation rate from a set of seven natural isolates. If mutation rate is a neutral trait, at mutation-drift equilibrium $VG=4NEVM$. The estimated VG for mutation rate is somewhere between 100X and 1000X below the predicted neutral rate. Further, since the mutation rate evolved both downward and upward, selection on mutation rate is evidently stabilizing and not uniformly directional (downward), as predicted by the drift barrier (but see below). In contrast, stabilizing selection is predicted by the leading (really the only) competing hypothesis of mutation rate evolution, the "cost of fidelity". Next, they extended that analysis to the between-species comparison with the sister species *S. paradoxus*. The squared divergence between species in mutation rate $D2=VM*T$, where T is the time of divergence. Estimates of *S. paradoxus* mutation rate and the divergence time are available, and the upshot is that the difference in mutation rate between the two species is 100-1000 times less than expected from neutral divergence – again, the signature of stabilizing selection.

The authors then draw on data about mutation rate, "effective genome size" and N_e from a diversity of species, using the same assumptions as Lynch (2011) used in his initial analysis, to reach the conclusion that the mutation rate is usually well above the drift barrier.

But wait, there's more! The authors then fish out four candidate genes, do some clever functional genomics, and identify one – PSP2 – as a "mutator" gene, defined here as a gene whose NORMAL function is to INCREASE the mutation rate.

OK, so the bottom line: I love this work. It is clever, elegant, addresses a question of fundamental importance in evolutionary biology, and provides a clear answer to that question. My only caveat with respect to the experimental work is this: I am not a microbiologist, and I am not competent to assess how rigorously generation time was estimated. Obviously, comparative studies of rates depend critically on what is in the denominator, i.e., the number of generations. I trust that one of the other reviewers IS competent to assess the rigor of the generation-time estimates, and that s/he will sign off.

We thank the reviewer for the careful review and praise of our work. We estimated the number of

generations between bottlenecks by counting the number of cells in an average-size colony and assuming exponential growth, and took the average of the estimates prior to and after MA. This is described in Methods and is the standard method used in yeast MA studies (e.g., see PMID 15611159, 18583475, and 29760081).

Now, a thought with respect to the interpretation. The majority of the second-order MA lines DID evolve a higher mutation rate, so on average, the mutation rate did increase with MA, i.e., under relaxed selection. As predicted by the DBH. Now, I do buy the stabilizing selection argument, but it seems to me that the fitness function is not symmetric around the current mean (or in other words, there is a mutational bias), which I interpret to mean that there is both a directional (linear) and quadratic (stabilizing) component to selection on mutation rate. Now of course the drift barrier may have nothing to do with this, but I would argue that the evidence suggests that the long-term outcome of evolution under relaxed selection would be an increased average mutation rate, and where it equilibrates would seem to depend on the effectiveness of selection. My own sentiment is that the cost of fidelity HAS to be true (sort of like synergistic epistasis HAS to be true!), and I think the results of this study strengthen the case. But still: relaxed selection, and mutation rate drifts upward, on average. I think the authors need to explicitly acknowledge that fact.

We completely agree that there is a mutational bias toward higher mutation rates in wild-type cells. That is, random mutations occurring in wild-type cells are more likely to increase the mutation rate than decrease the mutation rate. One piece of evidence for this view is that there had been ~30 reported “anti-mutator” genes in the yeast genome but no report of “mutator” genes prior to our study. Because of the mutational bias, the mutation rate will on average increase upon the removal of selection. So, the finding that most MA lines showed an increased mutation rate is consistent with both the drift-barrier hypothesis and the conventional model. However, the observation that ~30% of MA lines significantly reduced mutation rate after the removal of selection is consistent with the conventional model but inconsistent with the drift-barrier hypothesis. We have revised a sentence on page 5 to the following to clarify the issue.

“That 30% of MA lines with significantly altered μ exhibit such drastic reductions in μ is inconsistent with the DBH, because when μ is near the drift barrier, mutations are expected to be strongly biased toward increasing μ and are not expected to cause such large reductions of μ so frequently (Fig. 1b).”

And, I will note that this work has two historical antecedents. Way back in 2006, Ávila et al. (Genetics) did a second-order MA experiment in *Drosophila*, in which they reported that the mutation rate increased in the second-order MA lines (U estimated from fitness assays; I think their results are also consistent with synergistic epistasis, although they did not seem to think so). And in 2019, Saxena et al. (MBE) reported a second-order MA experiment with *C. elegans*, in which they observed that, on average, mutation rate increased under MA conditions, which they

argued was consistent with the DBH. They also reported mutational heritability for mutation rate on the order of $10^{-3}/\text{gen}$ (just like for every other trait under the sun).

Thanks for bringing these two papers to our attention, and they have now been cited on page 13.

A few minor comments, noted by Page #/Line #.

1. P6/L145. The term "phenotypic variance" is a bit confusing in this context. I know what you mean, which is the genetic variance for a phenotypic trait, but "phenotypic variance" usually connotes VP, not VG. Maybe think about how to rephrase.

Following a previous paper that used the same test (PMID: 15852004), we have rephrased "phenotypic variance" to "genetic variance among natural strains of a species (V_g)".

2. P8/L206. Your definition of "mutator" is different from usual. Again, the meaning is clear in context, but "mutator" alleles are typically considered to be alleles (presumably defective loss-of-function alleles) at what you call antimutator GENES that increase the mutation rate. This non-standard usage may lead to confusion down the road; an additional explanatory clause following the "30 antimutator genes" with respect to mutator alleles may smooth the path. Something to consider.

We have added the following sentence on page 8 to clarify this term.

"Note the distinction between mutator genes and mutator alleles, the latter being loss-of-function alleles of antimutator genes."